# Aligning Audio-Visual Joint Representations with an Agentic Workflow

**Shentong Mo**
CMU / MBZUAI
DAMO Academy, Alibaba Group
shentongmo@gmail.com

**Yibing Song**[*]
DAMO Academy, Alibaba Group
Hupan Laboratory
songyibing.syb@alibaba-inc.com

## Abstract

Visual content and accompanied audio signals naturally formulate a joint representation to improve audio-visual (AV) related applications. While studies develop various AV representation learning frameworks, the importance of AV data alignment is usually undermined for achieving high-quality representation. We observe that an audio signal may contain background noise interference. Also, non-synchronization may appear between audio and video streams. These non-strict data alignment limits representation quality and downgrade application performance. In this paper, we propose to improve AV joint representations from a data-centric perspective by aligning audio signals to visual data. Our alignment is conducted in an agentic workflow controlled by an LLM-based assistant named AVAgent. For each input AV data pair, our AVAgent uses a multi-modal LLM to convert audio and visual data into language descriptions separately (i.e., tool use). Then, AVAgent reasons whether this paired data is aligned well and plans to edit the audio signal if needed (i.e., planning). The audio editing is executed by predefined actions that filter noise or augment data. Moreover, we use a VLM to evaluate how modified audio signals match the visual content and provide feedback to AVAgent (i.e., reflection). The tool use, planning, and reflection steps operate cyclically to become an agentic workflow where audio signals are gradually aligned to visual content. To this end, existing methods can directly leverage the aligned AV data via our agentic workflow to improve AV joint representations. The experimental results comprehensively demonstrate the state-of-the-art performance of the proposed approach against previous baselines in diverse downstream tasks.

## 1 Introduction

Video stream is usually captured with sound recording. Intuitively, the audio signal supplements video data to formulate a joint AV representation. Compared to a single visual or audio representation, the joint representation benefits both single and cross-modality applications such as automatic captioning, content retrieval, and human-computer interaction. Learning audio-visual representations has been heavily investigated in the audio-visual recognition [1, 2, 3], sound source separation [4, 5, 6], and self-supervised form [7, 8]. These studies design various representation learning framework to leverage existing AV data pairs to obtain joint representations, which are further applied into downstream audio and visual related scenarios.

The audio and visual pairs may not align well in practice during data capturing. We observe there are two main issues. First, the audio signal may contain background noise interference. In the real capturing scene, the microphone may record sound unrelated to the visual content. Second, the recorded sound may not correspond to the video frames temporally. This may be because of the

---

[*]Y. Song is the corresponding author. The project page can be found at: https://avagent.github.io

38th Conference on Neural Information Processing Systems (NeurIPS 2024).

non-synchronization between capture and microphone. The audio may appear earlier or later than the frames. As a result, there appears misalignment between the AV data. A direct utilization of these data, although via the following designed learning framework, will still limit the AV joint representation quality.

In this paper, we improve AV joint representations from a data-centric perspective. Based on our observation that audio noise and non-synchronization lead to a non-strict AV alignment, we propose an agentic workflow to align audio signals to visual content adaptively. Fig. 1 shows an overview. Our workflow consists of tool use, planning, and reflection steps. These steps are controlled by an LLM-based agent named AVAgent. For one input AV paired data, our AV-

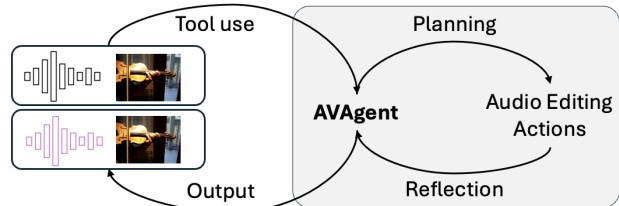

Figure 1: A glimpse of the AVAgent workflow. Three steps (tool use, planning, and reflection) form a cyclic agentic workflow where audio signals are progressively aligned with the visual content for joint representation improvement.

Agent will leverage a multi-modal LLM to convert audio and visual data into language descriptions, separately. Through this step, AVAgent independently perceives the audio and visual data for consistency judgment. Our AVAgent will reason based on the language descriptions and plan to edit audio signals if needed. The audio edition executes predefined actions that remove background noise (e.g., wiener filtering) and augment data (e.g., speed modification). Then, our AVAgent leverages a VLM to verify if the edited audio signal matches the video stream. This VLM provides feedback to the AVAgent for guidance in the next cycle. By tool use, planning, and reflection in this workflow, the audio signals are gradually aligned to the visual content to formulate an improved joint AV representation.

Our innovative use of LLMs in this data-centric workflow marks a significant advancement in the field of audio-visual representation learning. By focusing on improving data quality through intelligent audio editing, we establish a foundation for more accurate and robust audio-visual joint representations. We extensively evaluate our *AVAGENT* on Flick-SoundNet, VGG-Instruments, VGG-Music, VGGSound-All, and AVSBench datasets. The experimental results comprehensively demonstrate the state-of-the-art performance of the proposed approach against previous baselines in linear probing, fine-tuning classification, visual sound localization, sound separation, and audio-visual segmentation.

## 2   Related works

In this section, we provide an overview of the landscape of prior research in audio-visual representation learning and discuss how current innovations in the use of Large Language Models (LLMs) as agents contribute to advancements in this field.

**Audio-Visual Representation Learning**    Audio-visual representation learning has long been a focus of multimedia research [1, 2, 3, 9, 10, 4, 5, 6, 7, 8, 11, 12, 13, 14, 15, 16, 17, 18, 19, 20, 21, 22, 23], aiming to establish effective cross-modal correlations between audio and visual data. Pioneering works such as SoundNet [1] and the approaches by Owens et al. [2] and Arandjelovic et al. [3] have laid the foundation for understanding these modalities as intertwined rather than separate. These studies have shown that synchronizing audio with visual input enhances machine perception and can be pivotal for tasks such as event localization [24, 25, 26, 27, 28, 29, 30, 31, 32, 33, 34] and audio-visual spatialization [35, 36, 37]. Recent advancements have also explored complex scenarios like audio-visual navigation [37, 38, 39] and parsing [40, 41, 42, 43, 44], highlighting the depth and versatility of audio-visual data integration. Our focus on improving data quality through intelligent adjustments sets our work apart from existing methods, positioning it as a significant contribution to the field of AV representation learning.

**LLM-based Agents**    The integration of LLMs [45, 46] as decision-making agents represents a significant leap in multimedia processing. For instance, the AesopAgent [47] and VideoAgent projects [48] utilize LLMs to drive long-form video understanding and story-to-video production, showcasing the potential of LLMs in generating and refining multimedia content. However, these

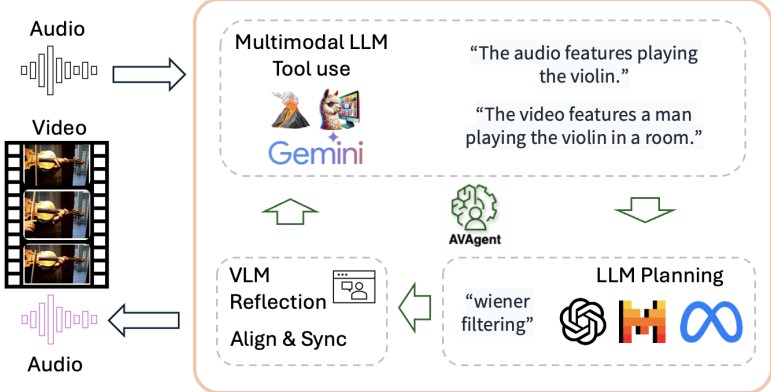

Figure 2: Overview of the *AVAGENT* framework. **Tool use**: For each audio-visual data pair, we employ a multi-modal Large Language Model (LLM) to convert audio and visual data into the language form, separately. **Planning**: The agent takes the AV data via text description and plans to edit the audio signal for alignment enhancement. **Reflection**: Subsequently, a Vision-Language Model (VLM) evaluates modifications to ensure that the audio adjustments appropriately match the visual content, and provides feedback to the agent. These steps form a cyclic agentic workflow where audio signals are progressively aligned with the visual content for enhanced joint representation.

applications primarily focus on generating or interpreting content rather than enhancing the quality of existing audio-visual pairings. In contrast to these works, our approach utilizes LLMs specifically tailored for each AV pairing to adaptively modify audio to better align with the corresponding video content. This method not only automates the process of audio editing but also ensures that the modifications are contextually appropriate, enhancing the overall coherence and synchronization between the audio and video streams. Our approach is particularly innovative in its use of Vision-Language Models and LLMs in conjunction to refine AV synchronization, a crucial aspect often overlooked in traditional methods. By leveraging the capabilities of LLMs for detailed, context-aware audio editing, we address the foundational issue of AV misalignment, thereby enhancing the efficacy and applicability of AV systems in a range of real-world scenarios.

## 3   Proposed method

We propose to enhance the alignment of audio-visual (AV) paired data for joint representation improvement. Our alignment is fulfilled in an agentic workflow where audio signals are gradually aligned to visual data. Figure 2 shows an overview of our workflow controlled by an LLM-based assistant named *AVAGENT* İn the following, we revisit the AV representation learning, illustrate our *AVAGENT* and analyze our aligned AV paired data.

### 3.1   Revisiting Audio-Visual Representation Learning

Audio-visual (AV) representation learning aims to fuse information from both audio and visual modalities to create a unified representation that captures the intrinsic correlations between these two streams. This integration is foundational for enhancing the performance of various multimedia applications, including speech recognition, event detection, and content-based retrieval. Traditional frameworks in AV representation learning, such as those introduced in SoundNet [1] and the works by Arandjelovic et al. [3], typically generate feature embeddings for each modality, which are then merged to form joint representations. The effectiveness of these representations hinges on the assumption that the paired audio and visual data are well-aligned and synchronized. In practice, AV representation learning models directly consume raw or minimally processed AV pair data to construct these joint representations. While effective, this direct approach generally overlooks the potential misalignments and asynchronies inherent in the source data. Misalignments, whether temporal or contextual, can degrade the quality of the learned representations, thereby reducing the overall performance of the system on downstream tasks.

**Audio Noise Filtering Actions:**

1.`Spectral Subtraction`: Removes background noise by estimating and subtracting the noise spectrum.

2.`Wiener Filtering`: Applies an optimal filter to minimize overall mean square error in noisy images or sounds.

3.`Wavelet Denoising`: Employs wavelet transforms to selectively remove noise while preserving essential details.

4.`Spectral Gating`: Implements a noise gate that only allows signals above a certain threshold to pass through, reducing noise.

**Audio Coordination Actions:**

5.`Speed Modification`: Changes the playback speed to synchronize audio with video motion.

6.`Pitch Modification`: Adjusts the audio pitch to match the visual content or correct mismatches.

7.`Volume Adjustment`: Modifies the volume dynamically to fit the video context or to simulate different listening environments.

8.`Filling Blanks`: Introduces synthetic elements in a controlled manner to blank audio to handle various acoustic environments effectively.

Figure 3: Audio editing action illustrations. We design 8 actions to edit audio signals for AV alignment. The first 4 actions are set to reduce background noise interference, and the last 4 actions are set to coordinate audio signals to visual data. Our *AVAGENT* plans to use these actions according to input AV data pairs.

Our method addresses these limitations by introducing a data-centric approach that first assesses and corrects any misalignment between the audio and visual data before they are used for representation learning. By ensuring that the input AV pairs are accurately aligned, our approach enhances the quality of the joint representations, thus improving the robustness and efficacy of downstream applications. This refinement step differentiates our method from the previous framework [17], which often assumes that the input data alignment is already optimal.

## 3.2 Agentic Workflow

We design an automatic workflow to adjust audio signals in accordance with visual data. It consists of tool use, planning, and reflection steps. Taking the raw AV pair data as input, our workflow still outputs AV pair data where the audio signal is well aligned with visual data. As we focus on data processing, our refined pair data can be widely utilized in various scenarios related to AV representation learning.

**Tool use** We leverage multi-modal LLMs [49, 50] to map audio and video streams into a unified space where both data can be described in the language form. We transform the audio and video separately to obtain two independent language descriptions, which our *AVAGENT* further analyzes for planning. The separate transformation benefits alignment identifications. For example, when a person is speaking in a noisy market, the video content depicts 'A person talking in a market' while the audio signal may be dominated by background noise interference. Such AV discrepancy will be reflected in separate transformations to language description while the discrepancy may not be noticed if transferred jointly. Meanwhile, transferring AV pair into language form will benefit our LLM-based *AVAGENT* to identify and plan actions accordingly.

**Planning** Our *AVAGENT* reasons the given AV pair data in language form and plans for the upcoming actions. We have predefined 8 actions in advance as shown in Figure 3. These actions are defined based on our observation that background noise interference and non-synchronization mostly limit the AV joint representations. As such, we introduce 4 noise-filtering operations to remove background interference, and 4 audio coordination operations to correspond audio signals to video streams. In the planning step, *AVAGENT* plans one action to execute. To train *AVAGENT* for action planning, we prepare paired data where there are video streams, audio signals, and actions. Each pair is annotated with a context description and a corresponding action that should be taken for better video and audio alignment. Our *AVAGENT* is based on Vicuna-v1.5-7b [51] and we adopt LoRA [52] tuning so as not to affect the original reasoning ability of LLM.

**Reflection** We evaluate how the modified audio signals match visual data after executing actions. The ImageBind [53] measures the similarity between different modalities of data, which we adopt to compute the AV alignment score and temporal synchronization score. These two scores provide feedback to the *AVAGENT* for planning in the next cycle. If our scores are relatively low, the action

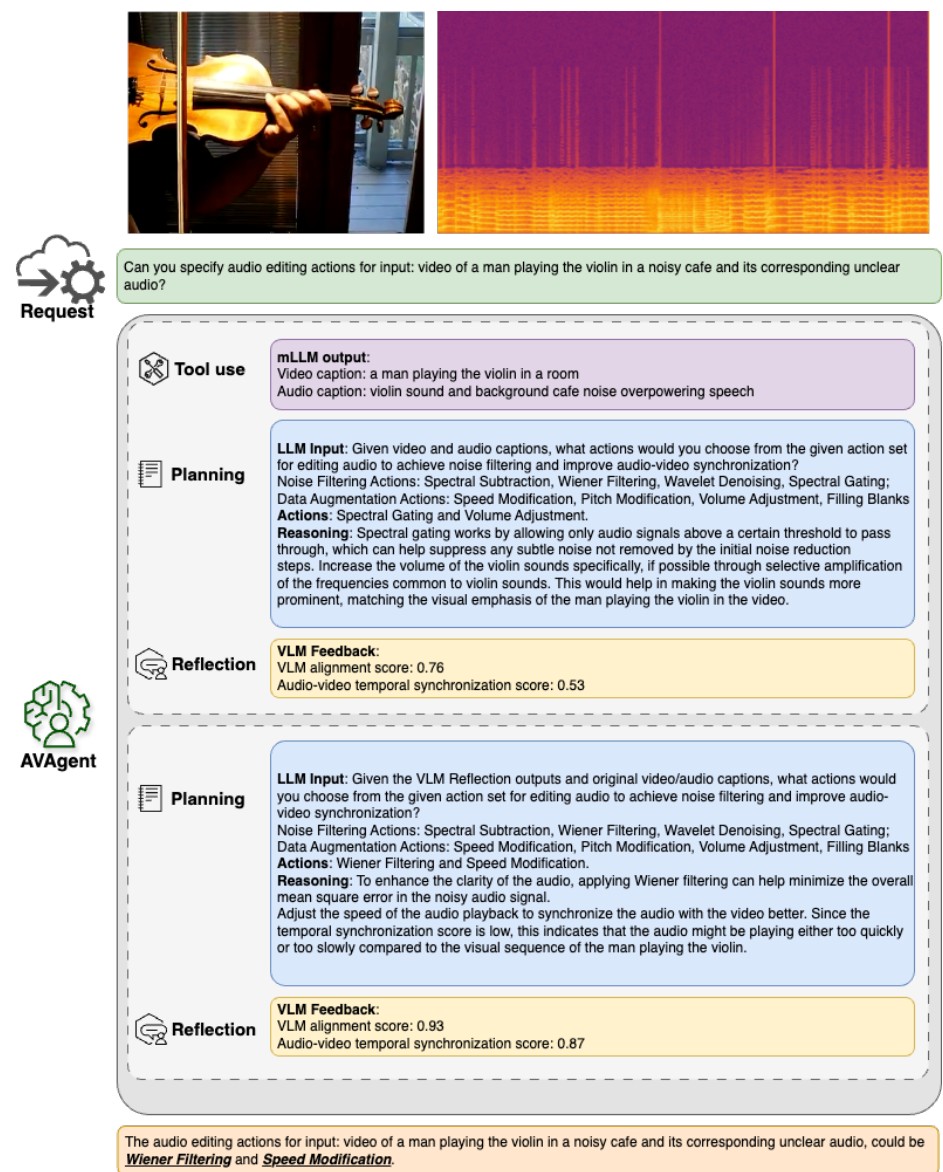

Figure 4: An example of our agentic workflow. For one input AV pair, we use mLLMs to transform video and audio data into language descriptions, separately. Then, AVAGENT reasons and plans for actions. After editing the audio, AVAGENT performs a reflection to compute two scores. As these scores are relatively low, they are sent to AVAGENT for consideration in the next cycle. The newly planned actions operate on the original input AV pair and achieve favorable scores in reflection. These actions are then identified for editing input audio signals.

planned in the next cycle will be put on the original AV pair, rather than the repaired AV data in the current round because of avoiding accumulated errors.

Figure 4 shows an example of how our workflow operates in practice. Given one input video and audio signals, our AVAGENT first uses mLLM to convert them into language descriptions, separately. Then AVAGENT reasons the input texts and plans two actions. One action is from the noise filtering aspect and another is from the audio coordination aspect. After modifying the audio signal, AVAGENT performs reflection by computing the alignment score and synchronization scores. These scores are relatively low at present and are sent to AVAGENT for consideration when planning actions. Then, two actions are selected to modify the original input AV data pair, followed by a reflection where the increased scores indicate the current two actions are suitable for aligning the input AV pair.

Table 1: **Data analysis of visual alignment and temporal synchronization.** True and False pairs denote the video corresponding to the modified and original audio separately.

(a) Visual alignment.

| True Pairs | False Pairs | Alignment (%, ↑) |
|---|---|---|
| 50k | 0 | **86.53** |
| 0 | 50k | 53.16 |
| 50k | 50k | 62.78 |
| 50k | 100k | 57.65 |
| 100k | 50k | 75.39 |

(b) Temporal synchronization.

| True Pairs | False Pairs | T-Alignment (%, ↑) |
|---|---|---|
| 50k | 0 | **78.23** |
| 0 | 50k | 42.05 |
| 50k | 50k | 52.29 |
| 50k | 100k | 45.65 |
| 100k | 50k | 63.71 |

## 3.3  Data Analysis

In this section, we present the data analysis procedures used to validate the effectiveness of our proposed method for improving audio-visual data alignment. Our approach involves quantitatively assessing the matching score between audio and video streams before and after applying our method, highlighting the significant enhancements achieved through our adaptive synchronization techniques.

To measure the alignment of audio-visual data, we introduce the visual alignment and temporal alignment score, which quantitatively evaluates how well the audio stream corresponds with the visual content in terms of timing and context auditory-visual congruence. The visual alignment and temporal alignment scores are computed using X-CLIP [54] that analyzes the coherence between the generated audio and the synchronization accuracy. Specifically, we perform simulation experiments by computing the metrics from modified (true) audio-video pairs and original (false) pairs.

**Visual Alignment**  For visual alignment, we compute the alignment (Alignment) score between video and audio features by using 50k modified (true) video-audio pairs and 50k randomly selected original (false) pairs from VGGSound [55]. The quantitative results are reported in Table 1a. With the increase in the number of false pairs with noisy visual information, all alignment metrics decrease. Adding 50k true pairs to [50k, 50k] cases further increases all metrics. These results validate the effectiveness of the proposed agent in removing audio data noises to improve visual alignment.

**Temporal Synchronization**  For temporal synchronization, we compute the temporal alignment (T-Alignment) score between video and audio features by using 50k modified (true) video-audio pairs and 50k randomly selected original (false) pairs averaged across all time steps. Table 1b shows the comparison results on our dataset in terms of T-Alignment scores. As can be seen, the T-Alignment score decreases with the increase in the number of false pairs without synchronized audio with the original videos, although they share the same visual information. Adding 50k additional true pairs to [50k, 50k] cases also increases T-Alignment scores, which further shows the importance of the proposed agent in increasing temporal synchronization together with visual alignment.

These analyses not only substantiate the effectiveness of our proposed adjustments but also illustrate the practical implications of our method in real-world scenarios. The improvement in matching scores post-intervention validates our approach, confirming that our method significantly enhances the synchronization and overall perceptual quality of audio-visual content. This section underscores the transformative potential of integrating large language models with multimodal large language models to achieve superior audio-visual data alignment, setting a new benchmark in the field of audio-visual representation learning.

## 4  Experiments

In this section, we provide the detailed experimental setup and evaluation protocols used to assess the performance of our proposed method on various audio-visual representation learning tasks. These experiments are designed to validate the effectiveness of our approach, highlighting its advantages over existing state-of-the-art methods.

### 4.1  Experimental Setup

Our experiments cover a range of audio-visual tasks, each chosen to demonstrate the robustness and versatility of our method in enhancing audio-visual data quality and alignment. The tasks

Table 2: **Audio-visual classification** on VGGSound-Music, VGGSound-All, and AudioSet datasets.

| Method | VGGSound-Music | | VGGSound-All | | AudioSet | |
|---|---|---|---|---|---|---|
| | Linear (%) | Finetune (%) | Linear (%) | Finetune (%) | Linear (%) | Finetune (%) |
| MAE [59] | 25.32 | 52.39 | 15.61 | 45.73 | 11.52 | 24.23 |
| AudioMAE [60] | 41.65 | 55.61 | 42.35 | 57.76 | 30.23 | 44.92 |
| CAV-MAE [61] | 60.53 | 67.26 | 55.27 | 65.53 | 40.56 | 51.29 |
| MAViL [62] | 61.95 | 69.53 | 57.36 | 67.17 | 43.62 | 53.38 |
| AV-MAE [63] | 60.82 | 67.61 | 56.15 | 65.08 | 41.67 | 51.32 |
| *AVAGENT* (ours) | **64.57** | **71.57** | **61.56** | **69.24** | **47.52** | **55.85** |

include audio-visual classification, audio-visual source localization, audio-visual segmentation, and audio-visual source separation.

**Datasets.** We utilize several well-known datasets in the audio-visual domain, including both synthetic and real-world scenarios, to ensure comprehensive testing conditions. These datasets are chosen to provide a variety of challenges in terms of noise levels, types of audio and visual content, and synchronization issues. Specifically, we use a subset of 144k pairs in VGG-Sound [55] for pre-training, and fine-tuning the model on audio-visual main downstream datasets. 1) For source separation, we used 40,908 video clips from 49 music categories for training and 1201 clips for testing, denoted as VGGSound-Music. VGGSound-Instruments [56] includes 32k video clips of 10s lengths from 36 musical instrument classes, a subset of VGG-Sound [55], and each video only has one single instrument class annotation. MUSIC [4] consists of 448 untrimmed YouTube music videos of solos and duets from 11 instrument categories, where We use 358 solo videos for training and 90 solo videos for evaluation. The used dataset is slightly smaller than the original MUSIC dataset since some videos are no longer publicly available to be downloaded. 2) For localization, we used Flickr-SoundNet [10] with 4,500 audio-visual pairs for training and testing the model on 250 audio-visual pairs of sounding objects and extended 250 non-sounding objects introduced in SLAVC [57]. 3) For audio-visual segmentation, AVSBench [58] includes 4,932 videos (in total 10,852 frames) from 23 categories, including instruments, humans, animals, etc. Following prior work [58], we used the same split of 3,452/740/740 videos for train/val/test. 4) For linear probing and fine-tuning, we applied VGGSound-Music with 49 classes and VGGSound-All with 221 categories for comprehensive evaluation.

**Evaluation Metrics.** Following the prior work [56, 64, 57], we use the Precision and F1 scores defined in [57] for visual source localization. For source separation, following [4], we use Signal-to-Distortion Ratio (SDR) and Signal-to-Artifact Ratio (SAR). For audio-visual segmentation, we apply mIoU and F1 scores as evaluation metrics, following the previous work [58]. Linear-prob and fine-tuning classification evaluations are based on top-1 accuracy, which measures the class difference from the ground-truth labels.

**Implementation.** Our models are implemented using state-of-the-art MAE framework [59] with specific optimizations to handle the large-scale data processing required for audio-visual tasks. Our method suggests modifications that are integrated into the preprocessing step of the learning pipeline, ensuring that the models are trained on high-quality, well-aligned audio-visual data. The input images are resized into a $224 \times 224$ resolution. The audio is represented by log spectrograms extracted from $3s$ of audio at a sample rate of 8000Hz. We follow the prior work [64] and apply STFT to generate an input tensor of size $128 \times 128$ (128 frequency bands over 128 timesteps) using 50ms windows with a hop size of 25ms. The models were trained for 100 epochs using the Adam optimizer [65] with a learning rate of $1e - 4$ and a batch size of 128.

## 4.2 Comparison to Prior Work

In this work, we propose a novel agentic workflow for aligning audio-visual joint representations. To demonstrate the effectiveness of the proposed *AVAGENT*, we comprehensively compare it to prior work on linear-prob/fine-tune, audio-visual sound source localization, separation, and segmentation. The results from these experiments are crucial in validating the effectiveness of our data-centric approach. By focusing on enhancing data quality through intelligent audio-visual synchronization, our method not only improves the accuracy of specific tasks but also enhances the overall utility of AV systems in diverse applications.

**Audio-visual classification.** To validate the effectiveness of the proposed *AVAGENT* on audio-visual classification, we compare to the following prior baselines: 1) MAE [59]: a masked autoencoder with

Table 3: **Sound source localization and segmentation.** Quantitative results on Flickr-SoundNet and AVSBench.

| Method | Flickr-SoundNet | | | AVSBench | |
| --- | --- | --- | --- | --- | --- |
| | Precision | AP | F1 | mIoU | F1 |
| Attention 10k [10] | 49.38 | 51.23 | 55.39 | 20.76 | 31.25 |
| OTS [66] | 51.23 | 53.28 | 58.12 | 24.55 | 36.85 |
| DMC [14] | 50.52 | 52.93 | 57.56 | 23.51 | 35.27 |
| CoarsetoFine [67] | 51.76 | 54.85 | 58.63 | 26.53 | 38.62 |
| DSOL [68] | 55.29 | 57.92 | 62.05 | 29.85 | 42.23 |
| LVS [69] | 52.38 | 55.31 | 59.35 | 27.32 | 40.18 |
| EZVSL [64] | 54.71 | 57.51 | 61.38 | 30.52 | 43.26 |
| Mix-and-Localize [56] | 55.83 | 58.21 | 62.52 | 31.69 | 45.35 |
| SLAVC [57] | 55.65 | 58.12 | 62.39 | 31.36 | 45.02 |
| *AVAGENT* (ours) | **58.23** | **60.76** | **65.03** | **36.37** | **49.85** |

Table 4: **Sound source separation.** Quantitative results on MUSIC and VGGSound datasets.

| Method | MUSIC | | VGGS-Instruments | | VGGS-Music | |
| --- | --- | --- | --- | --- | --- | --- |
| | SDR | SAR | SDR | SAR | SDR | SAR |
| NMF [70] | -0.62 | 2.41 | -3.85 | -0.76 | -7.12 | -9.01 |
| RPCA [71] | 0.86 | 3.81 | -2.39 | 1.58 | -5.53 | -7.82 |
| Sound-of-Pixels [4] | 4.55 | 10.24 | 2.52 | 4.67 | 0.95 | 1.03 |
| MP-Net [72] | 4.82 | 10.56 | 2.63 | 4.85 | 1.37 | 1.39 |
| CCoL [73] | 6.35 | 9.75 | 3.28 | 5.01 | 2.07 | 2.18 |
| OneAVM [17] | 7.38 | 7.48 | 5.36 | 5.52 | 2.51 | 2.61 |
| *AVAGENT* (ours) | **8.82** | **11.72** | **6.72** | **6.85** | **5.26** | **5.28** |

only images as input; 2) AudioMAE [59]: a masked autoencoder with only audio as input; 2) Audio-Visual MAEs [61, 62, 63]: masked autoencoders with both audio and images as input. Table 2 reports the quantitative comparison results in Table 2. As can be seen, we achieve the best performance in terms of linear probing and fine-tuning on two benchmarks. In particular, the proposed *AVAGENT* significantly outperforms MAE [59], the current impression masking modeling approaches on images, by 46.12% and 23.85% on VGGSound-All regarding linear probing and fine-tuning. Moreover, we achieve superior performance gains compared to AudioMAE [60] with masked modeling on only audio signals, which implies the importance of learning joint audio-visual representations for multi-modal recognition. Meanwhile, our *AVAGENT* outperforms MAViL [62], the state-of-the-art Audio-Visual MAE by a large margin, where we achieve the performance gains of 2.62%, 2.04% on VGGSoud-Music, 4.20%, 2.07% on VGGSoud-All and 3.90%, 2.47% on AudioSet. These significant improvements demonstrate the superiority of our method in audio-visual classification.

**Sound source localization and segmentation.** To validate the effectiveness of the proposed *AVAGENT* on sound source localization, we compare to the following prior work: 1) Attention 10k [10] (CVPR 2018): the first baseline on sound source localization using a two-stream and attention-based neural net; 2) OTS [66] (ECCV 2018): a correspondence-based baseline for localization; 3) DMC [14] (CVPR 2019): a deep multi-modal clustering approach based on audio-visual co-occurrences; 4) CoarsetoFine [67] (ECCV 2020): a two-stage approach using coarse-to-fine embeddings alignment; 5) DSOL [68] (NeurIPS 2020): a class-based method with two-stage training; 6) LVS [69] (CVPR 2021): a contrastive learning framework with hard negative mining to learn audio-visual correspondence maps; 7) EZ-VSL [64] (ECCV 2022): a recent weakly supervised localization framework based on multiple-instance contrastive learning; 8) Mix-and-Localize [56] (CVPR 2022): a recent method based on a contrastive random walk on a graph of images and separated sound sources. 9) SLAVC [57] (NeurIPS 2022): a strong baseline with momentum encoders and extreme visual dropout to identify negatives and solve significant overfitting. The comparison results are reported in Table 3. Compared to Mix-and-Localize [56], the current state-of-the-art multi-source localization baseline, we achieve the results gains of 2.40 Precision, 2.55 AP, and 2.51 F1 on VGGSound-Instruments. Furthermore, when evaluated on the challenging AVSBench benchmark, the proposed approach still outperforms Mix-and-Localize [56] by 4.68 mIoU and 4.50 F1. These results validate the effectiveness of our approach in learning discriminative cross-modal representations from audio and images for sound source localization and segmentation.

**Sound source separation.** To demonstrate the effectiveness of the proposed *AVAGENT* on source separation, we compare the following methods: 1) NMF [70]: a traditional signal processing approach based on non-negative matrix factorization to generate the spectrogram of each sound source; 2) RPCA [71]: a parameter-free baseline based on robust principal component analysis; 3) Sound-of-Pixels [4]: a deep learning approach that recovers separated audio conditioned on pixel-level visual

Table 5: **Comparison to random actions.** Quantitative results on VGGSound-All, Flickr-SoundNet, AVSBench, and VGGS-Music datasets.

| Method | VGGSound-All | | Flickr-SoundNet | | | AVSBench | | VGGS-Music | |
|---|---|---|---|---|---|---|---|---|---|
| | Linear (%) | Finetune (%) | Precision | AP | F1 | mIoU | F1 | SDR | SAR |
| Random Actions | 50.26 | 56.82 | 45.32 | 48.26 | 50.78 | 22.18 | 34.63 | 0.95 | 0.96 |
| *AVAGENT* (ours) | **61.56** | **69.24** | **58.23** | **60.76** | **65.03** | **36.37** | **49.85** | **5.26** | **5.28** |

features; 4) MP-Net [72]: an improved audio-visual method based on recursive separation from the mixture; 5) CCoL [73] (CVPR 2021): a cyclic co-learning framework based on sounding object visual grounding to separate individual sound sources. 6) OneAVM [17] (ICML 2023): a unified audio-visual framework for localization, separation, and recognition. We report the comparison results on three main benchmarks in Table 4. As can be seen, the proposed *AVAGENT* achieves the best performs the bestance in terms of all metrics on VGGSound-Instruments and VGGSound-Music. For example, on the challenging VGGSound-Music, we outperform OneAVM [17], the current state-of-the-art model with a unified audio-visual learning framework, by 2.75 SDR and 2.67 SAR. Meanwhile, we achieve the best results of SDR and competitive results of SAR on the MUSIC dataset. These improvements show the superiority of our method in sound source separation.

## 4.3 Experimental Analysis

In this subsection, we provide a detailed analysis of the experiments conducted to assess the effectiveness of our approach. The experimental results are analyzed to understand the impact of our method on audio-visual representation learning, particularly focusing on the improvements over naive approaches using random actions.

To demonstrate the effectiveness of LLMs in our *AVAGENT*, we further compare the performance of our method against a naive approach where random audio modifications are applied without the guidance of LLMs in Table 5. This comparison is crucial to demonstrate the necessity and efficiency of our intelligent, context-aware synchronization strategy. For the naive baseline, random actions such as arbitrary pitch adjustments, random noise addition, or unsystematic volume changes are applied to the audio-visual data. The audio-visual-visual data modified using our LLM-guided method significantly outperforms the randomly modified data in all evaluation metrics. For instance, in audio-visual classification tasks, our method achieves a higher accuracy rate, demonstrating the relevance and precision of the audio adjustments suggested by the LLM. The poor performance of the naive approach underscores the importance of precise, context-driven modifications, which are only possible through the understanding and analysis capabilities of our LLMs.

## 5 Conclusion

In this work, we introduced *AVAGENT*, a novel data-centric approach for enhancing audio-visual representation learning by utilizing Large Language Models (LLMs) as agents to achieve precise audio-visual synchronization and alignment. Our method diverges from traditional method-centric approaches, which primarily focus on algorithmic enhancements, and instead emphasizes the critical role of data quality in audio-visual learning processes. Our methodology leverages both Multimodal LLMs and advanced LLM techniques, including LoRA tuning, to analyze and adaptively modify the audio stream in alignment with the corresponding video content. Through a series of well-structured experiments across various audio-visual tasks such as classification, source localization, retrieval, question-answering, segmentation, and source separation, our approach has demonstrated significant improvements over existing methods. The success of our method confirms the importance of focusing on data quality and intelligent data manipulation in audio-visual representation learning. By ensuring that the audio and video streams are well-aligned and contextually synchronized, we can significantly enhance the effectiveness of audio-visual learning models, thereby improving their applicability in real-world scenarios.

**Limitations.** While our approach significantly advances the field of audio-visual representation learning, there are some limitations that merit further investigation. Our method's effectiveness is contingent on the initial quality of the audio and visual inputs. In scenarios where inputs are of poor quality or excessively noisy, the performance of our LLM and VLM might be compromised,

potentially leading to sub-optimal synchronization and alignment. Addressing these limitations will be crucial for enhancing the robustness and versatility of our approach.

**Broader Impact.** Our work sets a new benchmark in the field of audio-visual (AV) representation learning, demonstrating that a data-centric approach, powered by advanced AI models, can lead to substantial improvements in both the performance and utility of AV systems.

**Acknowledgement.** This work was supported by DAMO Academy through DAMO Academy Research Intern Program.

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

## Appendix

In this appendix, we provide the following material:

- addition implementation and datasets details in Section A,
- algorithm for our *AVAGENT* in Section B,
- additional experimental analyses in Section C,
- additional example results from our *AVAGENT* in Section D,
- additional discussions on limitations and broader impact in Section E.

## A   Implementation & Dataset Details

In this section, we provide more implementation and dataset details.

**Audio-visual classification.** For linear probing, we follow the prior work [59, 60] and extract frozen audio-visual representations from our *AVAGENT* pre-trained audio-visual masked autoencoder. Then we attach a linear layer as a head to the frozen features for training with the audio-visual classes. During training, we only fine-tune the linear head to evaluate the quality of pre-trained features. The models are trained for 50 epochs using the Adam optimizer [65] with a learning rate of $1e-4$ and a batch size of 128. For fine-tuning, we use the same optimizer and batch size settings, but all parameters are learnable.

**Sound source localization and segmentation.** For sound source localization, we train all baselines [64, 57, 56] using the same backbone (*i.e.*, ViT-Base) for audio/visual encoder with different proposed objectives in their original papers. The final localization map is generated through bilinear interpolation of the similarity map between audio/visual features from the last self-attention layer. The models are trained for 30 epochs using the Adam optimizer [65] with a learning rate of $1e-4$ and a batch size of 128. For segmentation, we follow the prior work [58], and apply an upsampling decoder on features from the last self-attention layer to generate the final segmentation mask. We use the binary cross entropy (BCE) loss between the prediction and ground-truth masks for training. The models are trained for 20 epochs using the Adam optimizer [65] with a learning rate of $1e-4$ and a batch size of 128.

**Sound source separation.** For sound source separation, we follow the previous method [4, 17] and attach an audio U-Net decoder to our pre-trained audio-visual encoders for separating sounds from the mixture. The decoder depth for self-attention layers is 8, and the decoder receives the representations of the audio mixture and the visual embeddings. We also apply multiple transposed convolutions and an output head to predict a time-frequency separation mask. This separation mask is then used to multiply the input mixture STFT to separate the audio. Similarly to [4], the target masks refer to the time-frequency bins where the source is the most dominant component in the mixture. The sound source separation is achieved by optimizing a binary cross-entropy loss over these binary targets. The model is trained for 20 epochs using the Adam optimizer [65] with a learning rate of $1e-4$ and a batch size of 128.

**Dataset Details.** We evaluated our method using several prominent audio-visual datasets:

- **Flick-SoundNet [10]:** a dataset consisting of natural soundscapes with associated Flickr images with 4,500 audio-visual pairs for training and testing the model on 250 audio-visual pairs of sounding objects and extended 250 non-sounding objects;
- **VGG-Instruments [56]:** contains video clips of musical instrument performances, with 32k video clips of 10s lengths from 36 musical instrument classes, a subset of VGG-Sound [55], and each video only has one single instrument class;
- **MUSIC [4]**: consists of 448 untrimmed YouTube music videos of solos and duets from 11 instrument categories;
- **VGG-Music [17]:** a dataset that features a collection of music videos with annotations related to the genre and instruments present;
- **VGGSound [55]:** a comprehensive dataset that includes a wide variety of sound categories and corresponding visual scenes, which contains categories, such as animals, instruments, vehicles, people, etc;

**Algorithm 1** Algorithm for *AVAGENT*

---

**Require:** A dataset of audio-visual pairs $\{(a_i, v_i)\}_{i=1}^N$
**Ensure:** Improved alignment and synchronization of audio-visual pairs
1: **for** each pair $(a_i, v_i)$ **do**
2:   Generate independent audio and visual captions using multimodal LLM:
3:   $c_i^a \leftarrow \text{mLLM}(a_i)$ {Audio caption}
4:   $c_i^v \leftarrow \text{mLLM}(v_i)$ {Visual caption}
5:   Assess reflection using VLM:
6:   $s_i \leftarrow \text{VLM}(a_i'', v_i)$ {Alignment score}
7:   $s_i^t \leftarrow \text{VLM}(a_i'', v_i)$ {Synchronization score}
8:   **while** alignment $s_i$ and synchronization $s_i^t$ exceeds threshold (0.85) **do**
9:     Plan audio modifications to increase $s_i$ and $_i^t$:
10:     $a_i' \leftarrow \text{LLM}(c_i^a, c_i^v, d_i)$
11:     Apply audio modifications:
12:     $a_i'' \leftarrow \text{ModifyAudio}(a_i', \text{actions})$
13:     Assess reflection using VLM:
14:     $s_i \leftarrow \text{VLM}(a_i'', v_i)$ {Alignment score}
15:     $s_i^t \leftarrow \text{VLM}(a_i'', v_i)$ {Synchronization score}
16:     **if** $s_i$ improved and $s_i^t$ improved **then**
17:       Update audio $a_i \leftarrow a_i''$
18:     **end if**
19:     Update alignment $s_i$ based on new score $s_i$
20:     Update synchronization $s_i^t$ based on new score $s_i^t$
21:   **end while**
22:   Store improved pair $(a_i, v_i)$
23: **end for**

---

- **AudioSet [55]:** a collection of 2,084,320 human-labeled 10-second sound clips drawn from YouTube videos with 632 audio event classes;
- **AVSBench [58]:** a benchmark for testing audio-visual synchronization and alignment in diverse settings, including 4,932 videos (in total 10,852 frames) from 23 categories, including instruments, humans, animals, etc.

## B  Algorithm for AVAgent

Our approach is formalized through the following algorithmic steps:

1. Input an audio-visual pair.
2. Use the multimodal LLM to independently generate text descriptions for both audio and visual data.
3. Analyze the descriptions to detect discrepancies and assess alignment using a trained LLM model.
4. Plan and apply necessary audio modifications to enhance alignment, such as noise filtering and temporal adjustments.
5. Use a VLM to assess the effectiveness of the modifications and provide a feedback score.
6. Iterate the process until the audio-video alignment meets the predefined threshold or maximizes the synchronization score.

Algorithm 1 outlines the steps followed by our *AVAGENT* in processing and synchronizing audio-visual data using a data-centric approach. The algorithm utilizes a multimodal large language model (mLLM), large language model (LLM), and a vision-language model (VLM) to refine audio to better align with the visual content iteratively. This algorithm encapsulates the cyclic process of audio signal refinement, leveraging both machine learning models and heuristic analysis to ensure that the audio accurately reflects the visual context. Each step of the process is designed to enhance the synchronization between the audio and visual modalities iteratively.

Table 6: **Ablation studies on LoRA tuning.** Quantitative results on VGGSound-All, Flickr-SoundNet, AVS-Bench, and VGGS-Music datasets.

| LoRA Tuning | VGGSound-All | | Flickr-SoundNet | | | AVSBench | | VGGS-Music | |
|---|---|---|---|---|---|---|---|---|---|
| | Linear (%) | Finetune (%) | Precision | AP | F1 | mIoU | F1 | SDR | SAR |
| ✗ | 59.87 | 67.95 | 56.93 | 59.37 | 63.86 | 33.79 | 46.58 | 3.89 | 3.92 |
| ✓ | **61.56** | **69.24** | **58.23** | **60.76** | **65.03** | **36.37** | **49.85** | **5.26** | **5.28** |

## C   Additional Experimental Analyses

In this section, we conduct additional experimental analyses to explore the impact of LLM (Low-Rank Adaptation) tuning in our `AVAGENT`. The results are reported in Table 6. LoRA tuning enhances our LLM's memory and context retention capabilities, enabling it to make more informed decisions regarding AV synchronization. We implement LoRA tuning on the LLM to fine-tune it specifically for tasks that require a deep understanding of the audio-visual context. This involves adjusting the LLM's parameters to remember better and integrate lengthy audio-visual sequences. The application of LoRA tuning leads to noticeable improvements in tasks such as audio-visual source localization and segmentation, where the precision of temporal and spatial alignments is crucial. Metrics such as mIoU and F1 score show substantial enhancements. These improvements can be attributed to the LLM's enhanced ability to maintain context over extended interactions, proving essential in complex scenarios where long-term dependencies are critical. These analyses highlight the efficacy of our adaptive, LLM-driven approach in achieving high-quality AV synchronization. By leveraging sophisticated LLM tuning methods and moving away from naive, random modifications, our method sets a new standard in AV data processing, paving the way for more accurate and effective multimedia applications.

## D   Additional Examples

In this section, we present detailed examples of the agent workflow in action, demonstrating the iterative process of aligning audio-visual content through the use of tool use, planning, and reflection phases. Each example includes inputs, model outputs, action plans, and reflection scores, showcasing the method's effectiveness in improving synchronization and alignment.

### D.1   Example 1: Lecture in a Large Hall

Figure 5 provides a full example of our agent workflow for lectures in a large hall.

### D.2   Example 2: Dog Barking in Park

Figure 6 provides a full example of our agent workflow for dog barking in the park.

### D.3   Example 3: Waterfall Scene

Figure 7 provides a full example of our agent workflow for the waterfall scene.

## E   More Discussions

### E.1   Limitations

Our approach, while advancing the state-of-the-art in audio-visual synchronization, is subject to several limitations that should be considered:

- **Dependency on Data Quality:** The performance of our method heavily relies on the quality of the input data. Poor quality video or audio, such as low-resolution video or highly distorted audio, can significantly impair the ability of the LLM and VLM to accurately generate useful captions and subsequent synchronization actions.

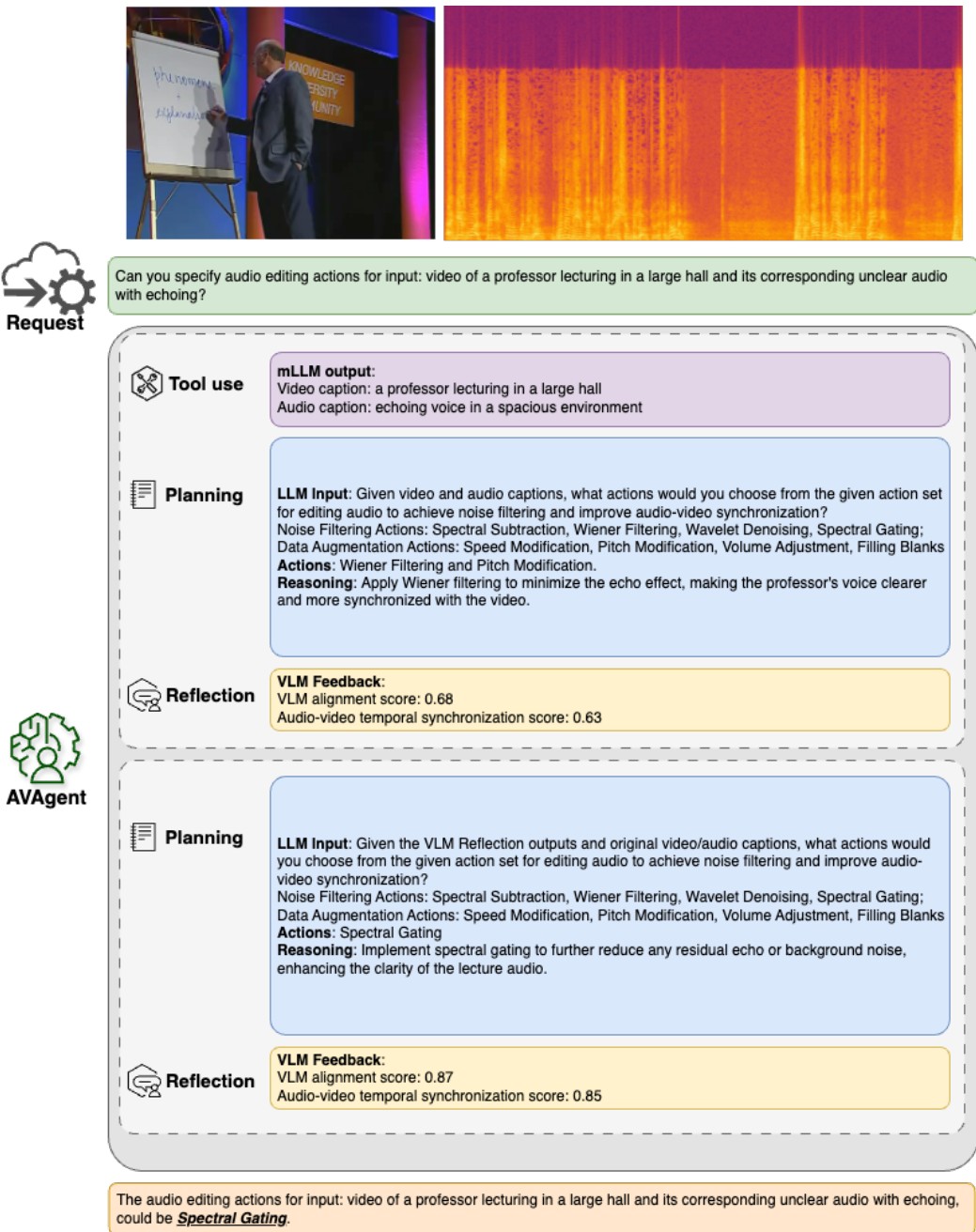

Figure 5: Illustration of a full example (Lecture in a Large Hall) of our agent workflow.

- **Computational Resources:** The algorithms employed, particularly the use of advanced LLMs like Vicuna-v1.5-7b and sophisticated VLMs, require substantial computational power. This may limit the practicality of our method for use in real-time applications or on platforms with limited processing capabilities.

- **Generalization across Diverse Environments:** While the method performs well across several tested environments and datasets, its effectiveness in extremely noisy or acoustically complex environments has not been extensively verified. There may be scenarios where the method's ability to discern and correct misalignments is diminished.

- **Scalability Issues:** The iterative nature of the workflow, while effective, may not scale efficiently with the volume of data or the complexity of the audio-visual scenes. This could

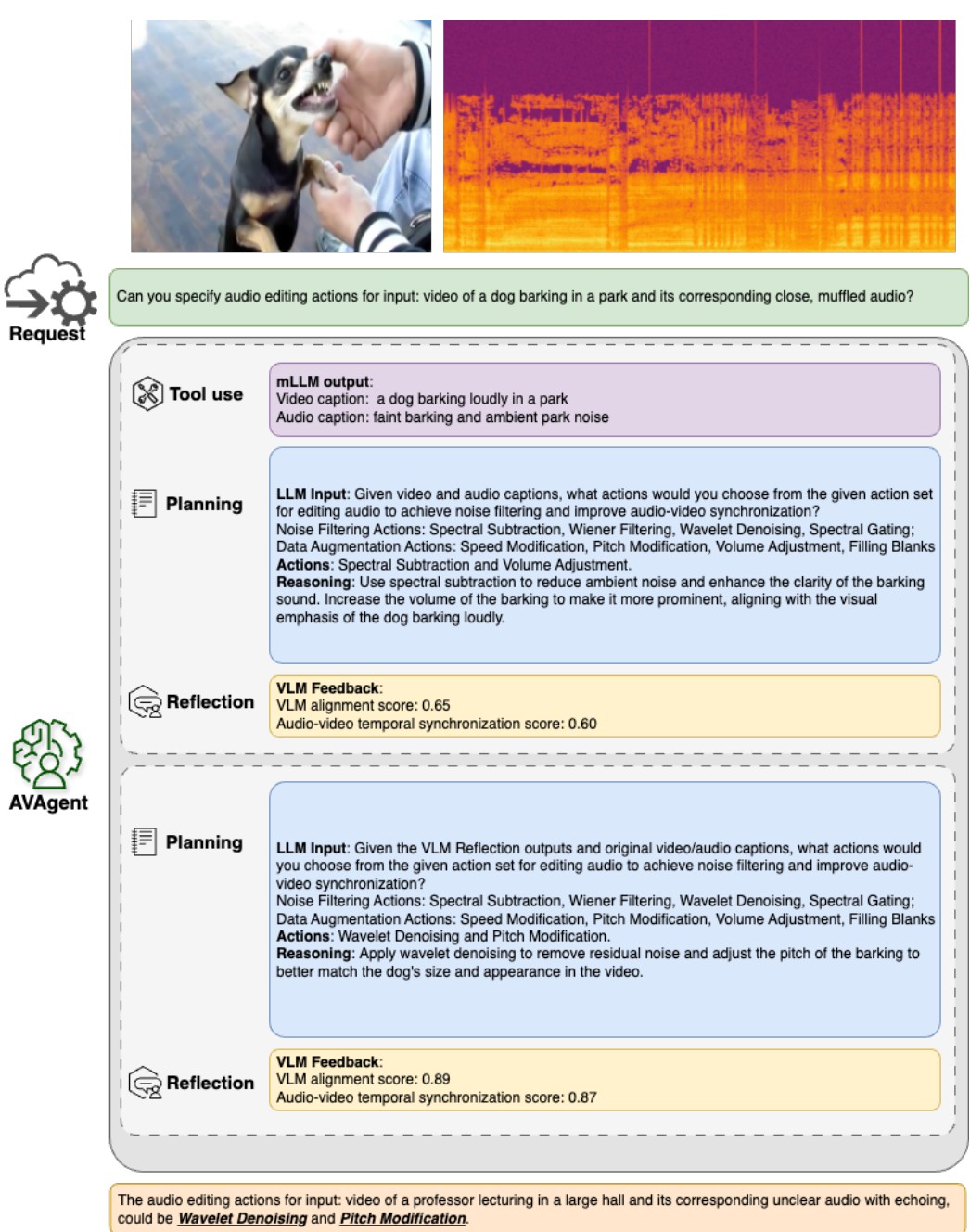

Figure 6: Illustration of a full example (Dog Barking in Park) of our agent workflow.

pose challenges in deploying the system at a larger scale, such as in industry-wide media processing pipelines.

- **Ethical and Privacy Concerns:** The use of LLMs and VLMs in processing potentially sensitive audio-visual content raises ethical and privacy concerns. There is a risk that such technology could be used to manipulate audio-visual content in misleading ways, or that sensitive information could be inadvertently exposed during processing.

## E.2   Broader Impact

The broader impact of our research is multifaceted, with potential implications for numerous fields:

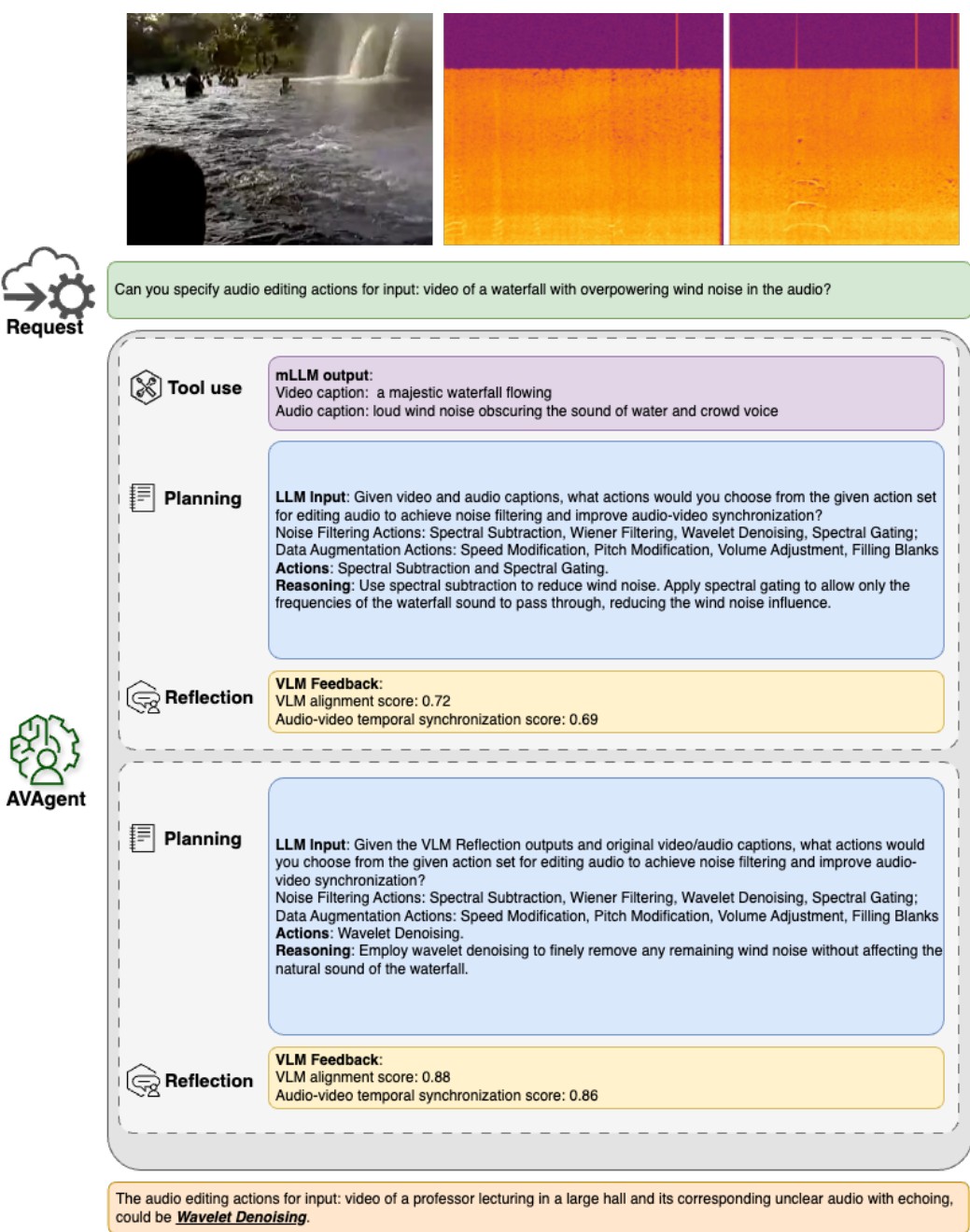

Figure 7: Illustration of a full example (Waterfall Scene) of our agent workflow.

- **Media and Entertainment:** Our method can significantly enhance the quality of multimedia content by ensuring better synchronization between audio and visual elements, leading to more immersive experiences in film, television, and virtual reality.

- **Accessibility Enhancements:** Improved audio-visual synchronization can also benefit accessibility technologies, such as developing more accurate closed captioning and audio descriptions for the hearing or visually impaired.

- **Educational Tools:** In educational settings, our method can be used to create more engaging and comprehensible instructional videos, where precise alignment of audio explanations with visual demonstrations is crucial.

- **Surveillance and Security:** Enhanced synchronization capabilities may improve the reliability of surveillance systems where audio cues are critical to understanding visual footage.
- **Ethical Considerations:** While our technology has many positive applications, it is essential to develop guidelines to prevent its misuse, particularly in contexts where audio-visual misalignment could be used to deceive or mislead viewers.

In conclusion, while our method presents significant advancements and potential benefits, it is essential to continue refining the technology to address its limitations and ensure it is used responsibly and ethically in diverse audio-visual applications.

