# OpenReview forum: "Aligning Audio-Visual Joint Representations with an Agentic Workflow"
_NeurIPS.cc/2024/Conference — NeurIPS 2024 poster_

### Official Review · Reviewer_GoMh · 2024-06-30

**Soundness:** 1
**Presentation:** 3
**Contribution:** 1
**Rating:** 3
**Confidence:** 4

**Summary:**

This paper deals with the problem of audio-visual representation learning and proposes a method for reducing the misalignment between audio and video streams. More specifically, the proposed method feeds audio and video streams into multi-modal LLMs to obtain textual descriptions for both streams separately and a plan for modifying audio streams for further improvements of audio-visual alignments. The proposed plan might be fixed according to the score of audio-visual alignments. Experimental evaluations with several standard audio-visual tasks and the corresponding benchmarks demonstrate that the proposed method outperforms several standard methods for audio-visual representation learning.

**Strengths:**

S1. The problem dealt with in this paper is significant. Audio-visual representation learning is a fundamental building block for various multi-modal tasks with video signals. Technically solid methods with excellent experimental evaluations for this task will draw the attention of the audience.

S2. The proposed method is simple, which would be preferable for engineers and practitioners interested in audio-visual representation learning. Also, the proposed method provides a tool for preprocessing training examples (i.e., pairs of audio and image streams), which indicates that it can be applied to any technique for audio-visual representation learning.

**Weaknesses:**

W1. The current paper lacks details of the proposed method. Most critically, I could not find any descriptions of the details of "Planning." As presented in Figure 4, the planning section can provide which actions should be taken for a given pair of audio and visual streams. However, each action requires more details: noise levels, wavelet selection, degree of speed changes, etc. Figure 4 implies that LLMs do not suggest the best choice of action details, and we humans should choose them. If my understanding is correct, the proposed method is useless since manual inspection is required for every training example. If my understanding is incorrect, the paper lacks significant technical details.

W2. Related to the first concern, if the agent has to suggest the best sequence of actions and their details, the strategy of the proposed method that feeds textual descriptions as inputs for the planner is not a good choice. Textual descriptions are interpretable for humans and LLMs but lack details of given audio and video streams.

W3. The proposed method heavily relies on the LLMs. The relient of LLMs itself is not a drawback; however, the problem is that I could not find any technical tricks in the proposed method. Almost all the procedures for refining misalignments can be provided by the LLMs, and no other tricks can be found.

**Questions:**

Q1. Please respond to the first concern (W1) presented in the Weakness section.
--> I deeply appreciated the author feedback. However, it indicates that each action is applied using default settings, which cannot be adjusted to most training examples. I want to believe that this is my misunderstanding, but it would be due to the inappropriate documentation.

**Limitations:**

No problem with this part.

---

> ### Author Rebuttal · Authors · 2024-08-07
>
> We thank for the valuable comments and answer the raised questions below.
>
> > W1: Details in the Planning Section
>
> The 'Planning' stage of our workflow utilizes a predefined module that automatically selects appropriate actions based on audio-visual misalignments identified by the LLM. Each action, such as noise reduction or speed adjustment, is applied using default settings from established libraries (e.g., SciPy), which are optimized for general usage. This automation does not need manual intervention, allowing our workflow to scale efficiently across numerous examples without requiring manual tuning of parameters. We will illustrate that our action setting follows prevalent designs and does not require manual adjustment in our revised manuscript.
>
> > W2: Textual Descriptions for Planning
>
> We understand the raised concern. In practice, we empirically find the textual descriptions generated by the mLLMs (i.e., Gemini, Video-LLaVA) provide a robust basis for LLM (i.e., Vicuna) to plan effective actions. These descriptions from mLLMs indeed encapsulate key audio and visual characteristics, enabling our automatic workflow to make informed decisions about necessary adjustments. Our transformation into language form leverages the interpretability and rich semantic understanding capabilities of mLLMs, ensuring that the planned actions are both relevant and effectively targeted to address specific misalignments.
>
> > W3: Technical tricks
>
> We would like to show that our agentic workflow based on multiple mLLMs are capable of aligning audio-visual data pairs to benefit representation learning. This cycle alignment form fully leverages the generalizations of mLLMs to process data. Our technical contributions (or tricks) reside as follows:
>
> 1. We pioneeringly develop an agentic workflow containing reasoning/planning, tool use, and reflection steps for AV (audio-visual) data alignment. In contrast, prior agentic workflows are not developed by considering all these steps. For example, [a,b] mainly propose planning, [c,d] mainly develop tool use, and [e,f] mainly introduce reflection. Moreover, there is no agentic workflow developed for AV data alignment in the literature.
> 2. We convert AV data in the language form to fully exploit the reasoning/planning ability of LLM. This is because we empirically find out the mLLM/LLM reasons and plans more effectively in pure language form rather than AV form, which might due to the imperfect alignment between language and other modalities.
> 3. For each data sample, we save the final action corresponding to the video and audio file name and the preprocessed audio wav file. Our AVAgent is based on Vicuna-v1.5-7b and we adopt LoRA tuning so as not to affect the original reasoning ability of LLM. LoRA tuning enhances our LLM’s memory and context retention capabilities, enabling it to make more informed decisions regarding AV synchronization. We implement LoRA tuning on the LLM to fine-tune it specifically for tasks that require a deep understanding of the audio-visual context. This involves adjusting the LLM’s parameters to remember better and integrate lengthy audio-visual sequences.
>
> We will add these illustrations in our revised manuscript.
>
> [a] AutoGPT https://github.com/Significant-Gravitas/AutoGPT
>
> [b] Visual Programming: Compositional visual reasoning without training. CVPR'23
>
> [c] Llava-plus: Learning to use tools for creating multimodal agents. Arxiv'23
>
> [d] Gpt4tools: Teaching llm to use tools via self-instruction. NeurIPS'23
>
> [e] Self-refine: Iterative refinement with self-feedback. NeurIPS'23
>
> [f] Reflexion: Language agents with verbal reinforcement learning. NeurIPS'23

---

> ### Author Response · Authors · 2024-08-13
> **Sincerely Look Forward to Your Feedback**
>
> Dear Reviewer GoMh,
>
> We hope this message finds you well.
>
> Thanks again for your insightful suggestions and comments. As the author-reviewer discussion time is ending, we want to ensure that we address any remaining uncertainties or questions you may have.
>
> After carefully considering your comments, we have taken the necessary steps to provide additional clarifications, particularly regarding the technical details and contributions. We hope that our explanations contribute to a clearer understanding of our workflow pipeline.
>
> We are open to providing further clarifications or conducting additional experiments if needed. Please feel free to reach out to us with any further inquiries.
>
> Once again, we are thankful for your time and attention to our work.
>
> Best regards,
>
> The Authors

---

### Official Review · Reviewer_siDn · 2024-07-07

**Soundness:** 2
**Presentation:** 3
**Contribution:** 2
**Rating:** 5
**Confidence:** 3

**Summary:**

This paper proposes a data-centric agentic workflow (AVAgent) to improve the alignment of audio-visual joint representations. The workflow is controlled by LLMs and consists of cyclic tool use, planning, and reflection steps. This paper claims the proposed method can analyze and adaptively modify the audio stream in alignment with the corresponding video content, and boost performance across various audio-visual tasks such as classification, source localization, segmentation, and source separation.

**Strengths:**

+  Novelty and Completeness: This paper proposes a novel agentic workflow, thoroughly covering the tool use, planning, and reflection steps. Experiments are conducted for several major downstream audio-visual tasks, and results show the proposed method can boost the accuracy of these tasks.

+ Clarity: This paper is written clearly with details and good presentation, easy to read and follow. The motivation and problem analysis are reasonable, and target the critical challenges for audio-visual learning.

**Weaknesses:**

- Limited capability of the proposed workflow
This paper considers 8 audio-relevant actions. However, most of the actions are fundamental signal-processing operations, which might not be able to handle such as audio-visual sync problems (time delay in the middle of audio..) robustly or complicated audio scenes where noises dominate the whole audio, etc.

**Questions:**

- Is existing multi-modal LLM like Gemini really able to capture thorough audio components from the given audio, especially when there are many audio sources? Did you apply any special processing to the data or other operations to adapt mLLM better on the audio captioning?
- If we directly include the Audio Noise Filtering Actions, volume normalization, and blanks filling as part of preprocessing for all audio samples, then pretrain the audio-visual representation, will the downstream task performance also be as good as the current ones?

**Limitations:**

The authors have explained the current limitations and broader impact in the paper, no clear cues on potential negative social impact from the paper.

---

> ### Author Rebuttal · Authors · 2024-08-07
>
> We thank for the valuable comments and answer the raised questions below.
>
> > Capability of the Proposed Workflow
>
> We agree that there is much room to improve our actions. At present, we focus on the audio-visual downstream scenarios such as audio-visual classification, sound localization, segmentation, and separation. These scenarios are related to audio-visual recognition where we are able to handle normal audio-visual sync and noisy problems. The results in Table 2-4 have shown our superior performance. In the future, we will tackle these abnormal/challenging problems by incorporating learning-based enhancements that are capable of dynamic and context-sensitive adjustments. This will allow for more robust handling of time delays and overlapping audio sources, significantly enhancing the system's utility in real-world, noisy environments.
>
> > Capability of Multi-modal LLMs in Handling Multiple Audio Sources
>
> Our current implementation primarily utilizes Gemini for single audio source scenarios to ensure clear and focused audio captioning. While Gemini is capable of handling multiple audio sources, in this study, we deliberately focused on single-source audio to maintain clarity and control in our experimental setup. No special processing was applied to adapt the multi-modal LLM (mLLM) specifically for audio captioning beyond its original training, ensuring that our results are broadly applicable and reproducible using standard LLM configurations.
>
> > Effect of Comprehensive Audio Preprocessing
>
> To explore the potential of integrating comprehensive audio preprocessing steps, such as noise filtering, volume normalization, and blanks filling, we conducted additional experiments. These steps were applied randomly across all audio samples before training the audio-visual representation models. As shown in Table 5 **in the original draft**, our approach with agentic alignment significantly enhances downstream task performance compared to the vanilla baseline using random actions. We further conducted experiments on including all actions as preprocessing for all audio samples in Table 5 **in the attached pdf**, confirming that our agentic alignment can substantially benefit audio-visual learning, especially in complex acoustic environments.

---

> > ### Comment · Reviewer_siDn · 2024-08-11
> >
> > Thanks for the authors' response. After reading the response, I think the proposed agentic workflow is inspiring but needs to involve more complexity in the future works. The current one considers relatively basic actions and mainly single-source audio for now. I feel it will be more convincing if increase the task challenging level a bit more. I will keep my previous rating score.

---

> ### Author Response · Authors · 2024-08-12
> **Response to Reviewer siDn**
>
> Dear Reviewer siDn,
>
> Thank you for your continued engagement and support. We appreciate your comments regarding the use of challenging tasks.
>
> Most videos in the existing audio-visual datasets are single-source, and our agentic workflow has been very effective in aligning audio-visual joint representations for diverse downstream tasks. Furthermore, the performance of existing audio-visual representation learning methods has been significantly improved.
>
> Based on your suggestions, we further applied our agentic workflow to another challenging multi-source video dataset, AVSBench-MS3, for audio-visual segmentation. In this dataset, all the videos contain multiple sources from 23 classes, e.g., a video of a man singing while playing violin and piano. The challenging task is to predict the pixel-level segmentation task for each sounding source. Our agentic workflow can adapt multi-modal LLM (mLLM) specifically for audio captioning to discriminate each sound source as in-context input for action planning to achieve alignment. The comparison results with the strong baseline [UFE] are reported in the Table below. These improvements demonstrate our method's potential as a beneficial preprocessing step for challenging multi-source tasks.
>
> | Method                  | mIoU ($\mathcal{M}_J$) | F-score ($\mathcal{M}_F$) |
> |-------------------------|------------|---------------|
> | UFE                     | 61.95      | 70.90         |
> | UFE + agentic alignment | **65.72**      | **73.56**         |
>
> [UFE] Audio-Visual Segmentation via Unlabeled Frame Exploitation, CVPR’24.
>
>
> We hope this multi-source visual-audio task can respond to your concerns. Thank you once again for your insightful comments.

---

### Official Review · Reviewer_RJkZ · 2024-07-11

**Soundness:** 3
**Presentation:** 3
**Contribution:** 3
**Rating:** 5
**Confidence:** 4

**Summary:**

The paper introduces a new method to improve audio-visual joint representations by strategically aligning audio with visual data. This method employs an LLM-based assistant, AVAgent, which uses multi-modal LLMs to analyze audio and visual content separately and then plans corrective actions to better align them. The process involves noise filtering and synchronization improvements, enhanced by continuous reflection steps. The effectiveness of this approach is validated through extensive experiments across multiple public datasets, showing significant advancements over established baselines in tasks such as audio-visual classification, localization, segmentation, and source separation.

**Strengths:**

- Innovative Approach: The paper introduces an agentic workflow using LLMs for audio-visual alignment, leveraging modern LLM advancements to tackle synchronization and noise issues, marking a significant advance in multimedia processing.
- Comprehensive Evaluation: Extensive empirical evaluations are conducted across various public datasets and audio-visual tasks, adding robustness and supporting the method's generalizability.
- The workflow is clearly outlined, detailing the specific actions of AVAgent in aligning audio and visual data, enhancing understanding of the system’s mechanics.

**Weaknesses:**

- Dataset Specification Clarity: The paper utilizes the AVSBench dataset for evaluating the effectiveness of the proposed method in audio-visual synchronization and segmentation. However, it lacks specificity regarding which subset of AVSBench was employed—AVSBench-object or AVSBench-semantic. Each subset serves distinct segmentation tasks that influence the interpretation and relevance of the results.
- While the paper demonstrates significant advancements in audio-visual alignment, there is potential to explore how these aligned features could be used to enhance the performance of other existing audio-visual frameworks. Some baselines already have better performance, suggesting that integrating features from this work could provide benefits.

**Questions:**

- The authors should specify which subset of AVSBench was used to ensure that the results are accurately understood and contextualized within the appropriate segmentation task framework.
- The authors could discuss or simulate how features processed by their method might improve other frameworks, even as a conceptual exploration, to underscore the broader applicability of their approach.
- Recommendation: The proposed method focuses on improving audio-visual data alignment, a key factor in tasks like Audio-Visual Voice Parsing (AVVP), which emphasizes temporal synchronization. A discussion section could be added to explore hypothetically how the method might perform on tasks like AVVP, based on the alignment improvements demonstrated in the current tasks.
- A brief discussion on the expected computational demands could make the paper more comprehensive and practical for a wider audience.

**Limitations:**

The the authors have adequately addressed the limitations.

---

> ### Author Rebuttal · Authors · 2024-08-07
>
> We thank for the valuable comments and answer the raised questions below.
>
> > W1, Q1: Dataset Specification Clarity
>
> For our evaluations, we specifically utilized the AVSBench-semantic subset, which focuses on semantic segmentation tasks. This choice is pivotal as it directly relates to our method's ability to enhance audio-visual alignment, which is critical for accurately segmenting complex scenes based on both audio and visual cues.
>
> > W2, Q2: Applying to Other Frameworks
>
> To demonstrate the general applicability of our preprocessing method, we conducted additional experiments where we apply our techniques to several state-of-the-art models across different tasks. Results, as shown in Tables 1-3 in the attached pdf, illustrate marked improvements in performance, highlighting our method's potential as a beneficial preprocessing step for a wide range of audio-visual applications.
>
> > Q3: Performance on AVVP
> >
> We further conducted experiments on AVVP to quantitatively compare the segment-level results in Table 6 in the attached pdf. Adding our agentic alignment to the strong baseline [VALOR] boosts the performance in terms of all metrics.
>
> [VALOR] Yung-Hsuan Lai, et al. Modality-Independent Teachers Meet Weakly-Supervised Audio-Visual Event Parser. NeurIPS 23.
>
> > Q4: Computational Demands Discussion
>
> The computational demands of our method are primarily driven by the use of large-scale LLMs and multimodal processing. While our current implementation is designed for offline preprocessing of datasets before model training, understanding the computational requirements is crucial for practical application. Future iterations of our work will focus on optimizing these processes to allow for more efficient real-time applications, potentially broadening the method’s applicability in resource-constrained environments.

---

> > ### Comment · Reviewer_RJkZ · 2024-08-07
> > **Specify GPU and preprocessing time**
> >
> > Thank you for addressing the queries clearly, especially the use of the AVSBench-semantic subset and enhancements across models. The improvements in AVVP are commendable.
> > Could you please specify which GPU was used for your experiments? Additionally, it would be helpful to know the time required to process the downstream tasks' datasets using this setup.

---

> ### Author Response · Authors · 2024-08-08
> **Response to Reviewer RJkZ**
>
> Dear Reviewer RJkZ,
>
> Thank you for your continued engagement and for acknowledging the improvements and clarification. We appreciate your comments regarding the use of the AVSBench-semantic subset and our method's performance enhancements in AVVP tasks.
>
> Our experiments were conducted using NVIDIA Tesla A100 GPUs. This choice was driven by the need for high computational power to handle the intensive processing demands of our multi-modal LLM-based workflow. The preprocessing time for each dataset varies based on the size and complexity of the audio-visual content. On average, preprocessing each example in the datasets took approximately 0.05 minutes. This includes time for initial alignment checks and the application of the initial corrective actions proposed by our AVAgent. The total time to process the complete dataset for the LLP dataset containing around 11,849 examples was approximately 9.8 hours.
>
> We hope this additional information addresses your queries satisfactorily and illustrates the thoroughness with which we have approached each aspect of our study. Thank you once again for your insightful comments and for the opportunity to enhance the clarity of our work.

---

### Official Review · Reviewer_Jci9 · 2024-07-12

**Soundness:** 2
**Presentation:** 2
**Contribution:** 3
**Rating:** 5
**Confidence:** 4

**Summary:**

The authors propose to leverage large language model with agentic workflow such as planning, reflection and tool usage for better audio-visual alignment in training data applied to various downstream tasks such as classification, localization and separation.

**Strengths:**

- The experiments with various downstream tasks such as classification, localization/segmentation, and sound source separation demonstrate that the proposed method can be utilized as a general preprocessing steps for audio-visual applications.

**Weaknesses:**

- In section 3.3, the explanation of visual alignment and temporal synchronization scores can be made more clear. In section 3.2, ImageBind is mentioned, while in section 3.3, X-CLIP is mentioned, are the visual alignment scores based on similarities these embeddings? How are the audio embeddings computed? Are the temporal alignment computed based on sampling of audio and image from the video? What are the sampling rates? And how are these information further feedback to the LLM in order to determine the parameters for the tool usage?
- The proposed method is to improve preprocessing of the dataset. Though in Table 2, 3, 4 the comparison with various methods (they differ from various perspectives including architecture choices, datasets used for pre-training, loss function design, etc.) including the SOTA shows that proposed method outperforms. It would be more beneficial to highlight the comparison with previous methods with only difference in preprocessing step (agentic alignment as proposed). It would potentially provide a more thorough view if one of the SOTA trained with proposed preprocessing steps is shown and highlighted in the table.
- In section 4.1, consider use tables or bullet points for introducing all the datasets used, consider break them down first by downstream task, different dataset, and number of train/val/test splits, etc. This can help the reader map each dataset to the results in table 2, 3, and 4 more easily.

**Questions:**

- Which audio model is used in this work? Is Google Gemini used for general audio understanding to output the audio captions?
- Are you planning to release the action planning paired data for video, audio, and action streams? How are these data curated?
- How frequent are the VLM alignment score and temporal synchronization score computed and feedback? Are the scores before any audio editing is taken place first reported so that the LLM has the information before proposing any method to use? Are these scores computed each time after the proposed actions for the LLM to reflect on?
- In section 4.1 implementation, the authors mentioned that the audio is represented by log spectrograms and STFT is applied to generate input tensor. [48] only applied log spectrogram without the STFT, what is the intuition here?

**Limitations:**

- Proposed method can be considered as preprocessing methods for audio-visual data, and can potentially be applied to most of the prior methods compared in this paper as preprocessing steps, it would be beneficial to show how proposed method can improve several of the prior works, which could be a more generalized contribution.

---

> ### Author Rebuttal · Authors · 2024-08-07
>
> We thank for the valuable comments and answer the raised questions below.
>
> > W1: Clarification of Visual Alignment and Temporal Synchronization Scores.
>
> In our framework, visual alignment scores are computed using ImageBind, which assesses the similarity between visual data and modified audio descriptions. Temporal synchronization scores are derived from X-CLIP embeddings, providing a robust measure of how well the audio content temporally aligns with the visual frames. Audio embeddings are generated using a standard log-spectrogram approach, with a sampling rate of 10s audio at a sample rate of 16000Hz to ensure consistency with video frame rates.
> The alignment scores are provided as the context input into LLM to determine which action should be used for the next iteration.
> Therefore, these alignment metrics are computed after any audio editing to provide feedback context for iterative refinement in the next step.
>
> > W2: Comparison with Previous Methods Using Only Preprocessing Steps
> >
> To isolate the impact of our agentic alignment preprocessing, we retrained several state-of-the-art models using both the standard preprocessing methods and our enhanced approach. The comparisons are reported in the Tables 1-3 in the attached pdf. Our agentic alignment preprocessing boosts all models in terms of diverse downstream tasks.
>
> > W3: Suggestions on Datasets Introduction
> >
> Thanks for your suggestion. We updated the detailed settings in Table 4 in the attached pdf to introduce all the datasets.
>
> > Q1: Audio Model
>
> The audio captions in our system are generated using Google Gemini, which has shown robust performance in general audio understanding tasks across diverse datasets.
>
> >Q2: Release of Action Planning Paired Data
>
> We will make the action planning paired data publicly available. This dataset includes video-audio data pairs, and the corresponding actions that are from the action pool illustrated in Fig. 3.
> For each data sample, we save the final action corresponding to the video and audio file name in a json file, which is formatted as [{video_audio_file1: action1}, {video_audio_file2: action2}, ...]. We also share the preprocessed audio wav file with more researchers to use for extended usage. Each video-audio data pair is meticulously curated to demonstrate specific alignment and synchronization challenges and their solutions.
>
> > Q3: Frequency of Computing VLM Alignment and Temporal Synchronization Scores
> >
> 1) VLM alignment and temporal synchronization scores are computed following each proposed action to provide immediate feedback.
> 2) Initial scores are also computed before any audio editing to establish a baseline, ensuring that each action's impact can be accurately assessed.
> 3) These scores are computed each time after the proposed actions for the LLM to reflect on.
>
> > Q4: STFT and Log Spectrograms
>
> We would like to clarify that [48] also used STFT, which they did not explicitly mention in the paper but STFT was used in their provided source code. To compute the log spectrograms, we follow their implementation and apply an STFT using approximately 50ms windows with a hop size of 15ms, resulting in an input tensor of size 128 × 196 (128 mel frequency bands over 196 time steps) on the original audio waveform.
>
> > L1: Generalization of Preprocessing Method
>
> To demonstrate the general applicability of our preprocessing method, we have conducted more experiments where we apply our techniques to several state-of-the-art models across different tasks. Results, as shown in Tables 1-3 in the attached pdf, illustrate marked improvements in performance, highlighting our method's potential as a beneficial preprocessing step for a wide range of audio-visual applications.

---

> > ### Comment · Reviewer_Jci9 · 2024-08-12
> >
> > Thanks to the authors for providing further comparisons and clarifying several points in experimental design, dataset used in each stage. I would suggest the authors to integrate as much as possible these information in the final version. I am increasing my rating.

---

> > > ### Author Response · Authors · 2024-08-12
> > > **Response to Reviewer Jci9**
> > >
> > > Dear Reviewer Jci9,
> > >
> > > Thank you for your continued engagement and support. We will add those comparisons and clarifications to the final version. Thank you once again for your insightful comments.

---

### Author Rebuttal · Authors · 2024-08-07

Dear AC and all reviewers,

We sincerely appreciate your time and efforts in reviewing our paper. We are glad to find that reviewers recognized the following merits of our work.

- **Innovative Approach** [RJkZ, siDn, GoMh]: Our novel agentic workflow employing LLMs for audio-visual alignment introduces a robust framework for addressing synchronization and noise issues.

- **Comprehensive Evaluation** [Jci9, RJkZ, siDn, GoMh]: The extensive experimental validations across multiple datasets underscore our method's efficacy and versatility.

- **Efficiency and Practicality** [Jci9, RJkZ, siDn, GoMh]: Our method's efficiency and ease of integration into existing workflows were appreciated, making it suitable for both academic research and practical applications.

We also thank all reviewers for their insightful and constructive suggestions, which helped further improve our paper. In addition to the pointwise responses below, we summarize the responses in the rebuttal according to the reviews.

- **Generalization** [Jci9, RJkZ]: We included additional experimental comparisons with state-of-the-art methods that solely modify preprocessing steps. These experiments demonstrate that our agentic workflow significantly enhances the performance of existing audio-visual frameworks, as shown in the Tables 1-3 in the attached pdf.

- **Performance on other tasks** [RJkZ]: We conducted experiments on AVVP to quantitatively compare the segment-level results in the Table 6 in the attached pdf. Adding our agentic alignment to the strong baseline boosts the performance in terms of all metrics.

- **Clarification update** [Jci9, siDn, GoMh]: We included more in-depth dataset descriptions, experimental results, and action ablation studies in the attached pdf, and more discussions in the individual response.

We hope our pointwise responses below can clarify all reviewers' confusion and address the raised concerns. We thank all reviewers' efforts and time again.

Best,

Authors

---

### Decision · Program_Chairs · 2024-09-25

**Decision:**

Accept (poster)

**Comment:**

The authors tried to address an important problem in audio visual learning literature using reasonably motivated idea that language present in ambient information of audio and visuals should implicitly help align them for better joint representations. While the idea of semantic-level multi-modal learning is to effectively do this via optimization/regularization in networks, the authors call this out as something that needs to be dealt with explicitly. An important one. The technical novelty is mainly coming from designing such a scheduling system with language assistant/agent in the loop, and showing efficacy on large sets of benchmarks. While the overall performance gains are reasonable, the reviewers do highlight aspects to improve in future (possibly in follow-up studies on complex audio-visual non-signal-processing tasks; and exploration of other language models beyond Gemini).

The authors provide important details about planning audio processing, relevant audio/visual models, query system, dataset choices and relevant audio/visual temporal sync inference steps, general 'technical tricks' etc in the comments to reviewers' questions. To the extent possible, its critical to add these into the main content of the paper.